# Predicting Respiratory Rate from Electrocardiogram and Photoplethysmogram Using a Transformer-Based Model

**DOI:** 10.3390/bioengineering10091024

**Published:** 2023-08-30

**Authors:** Qi Zhao, Fang Liu, Yide Song, Xiaoya Fan, Yu Wang, Yudong Yao, Qian Mao, Zheng Zhao

**Affiliations:** 1School of Medicine and Biological Information Engineering, Northeastern University, Shenyang 110819, China; zhaoqi@mail.neu.edu.cn (Q.Z.); wswy871107@gmail.com (Y.W.); 2School of Information Technology, Dalian Maritime University, Dalian 116026, China; fangliu@dlmu.edu.cn (F.L.); songyide@dlmu.edu.cn (Y.S.); 3School of Software, Key Laboratory for Ubiquitous Network and Service Software, Dalian University of Technology, Dalian 116024, China; xiaoyafan@dlut.edu.cn; 4Department of Electrical and Computer Engineering, Stevens Institute of Technology, Hoboken, NJ 07030, USA; yyao@stevens.edu; 5School of Light Industry, Liaoning University, Shenyang 110136, China; 6School of Artificial Intelligence, Dalian Maritime University, Dalian 116026, China

**Keywords:** deep learning, ECG, PPG, respiratory rate prediction, transformer

## Abstract

The respiratory rate (RR) serves as a critical physiological parameter in the context of both diagnostic and prognostic evaluations. Due to the challenges of direct measurement, RR is still predominantly measured through the traditional manual counting-breaths method in clinic practice. Numerous algorithms and machine learning models have been developed to predict RR using physiological signals, such as electrocardiogram (ECG) or/and photoplethysmogram (PPG) signals. Yet, the accuracy of these existing methods on available datasets remains limited, and their prediction on new data is also unsatisfactory for actual clinical applications. In this paper, we proposed an enhanced Transformer model with inception blocks for predicting RR based on both ECG and PPG signals. To evaluate the generalization capability on new data, our model was trained and tested using subject-level ten-fold cross-validation using data from both BIDMC and CapnoBase datasets. On the test set, our model achieved superior performance over five popular deep-learning-based methods with mean absolute error (1.2) decreased by 36.5% and correlation coefficient (0.85) increased by 84.8% compared to the best results of these models. In addition, we also proposed a new pipeline to preprocess ECG and PPG signals to improve model performance. We believe that the development of the TransRR model is expected to further expedite the clinical implementation of automatic RR estimation.

## 1. Introduction

Defined as the number of breaths in a minute (b/m), the respiratory rate (RR) serves as a robust indicator of an individual’s physiological state and is also a critical indicator of any physiological deterioration [1,2,3]. It has been advocated as one of the most pivotal early warning signals for cardiopulmonary arrests [1,2,3,4], implicated in conditions such as pneumonia, pulmonary embolism, hypercarbia, sepsis, and COVID-19 [5].

Currently, RR is extensively utilized in emergency departments in hospitals as a warning signal. It is largely measured using a labor-intensive manual counting-breaths method, which is unsuitable for sustained RR monitoring. To address this, several wearable sensors, such as impedance plethysmography (IP) sensors [6] and inductance plethysmography sensors [7], have been developed in recent years, paving the way for real-time RR monitoring. However, these sensors necessitate the placement of a tight band around a patient’s thorax, often resulting in discomfort during extended use [8]. Another type of sensor measures RR by detecting respiratory airflow or the carbon dioxide concentration [9]. Unfortunately, these sensors tend to interfere with normal breathing during RR monitoring [10].

In the recent decades, numerous methods have been proposed to predict RR from electrocardiogram (ECG) [11] or photoplethysmogram (PPG) [12] signals (or both), thereby creating opportunities for electronic, automatic, and unobtrusive of RR measurement in both fitness monitoring and healthcare contexts. ECG and PPG signals are physiological signals generated by the heart and the circulatory system and exhibit a significant correlation with respiration. During breathing, the expansion and contraction of the lungs lead to chest movements, which in turn affect heart movement and the state of blood vessels, resulting in alterations to ECG and PPG signals [13,14]. As such, ECG and PPG signals encapsulate the information about breathing frequency. Moreover, due to their non-invasive nature and widespread adoption in medical and health monitoring domains, ECG and PPG signals can be readily obtained.

Initially, most RR prediction methods utilizing ECG and PPG signals were based on a signal-analysis approach. The first algorithms for the estimation of RR from ECG and PPG signals emerged in 1985 and 1992, respectively [15,16]. Since then, a multitude of methods following this same approach have been proposed, with these methods receiving thorough comparisons in the literature [17]. Such methods typically extract respiratory-related signals from raw ECG or PPG signals (or both), including baseline wander (BW), amplitude modulation (AM), and frequency modulation (FM). Key features are extracted from the respiratory-related signals via various signal processing methods in the time domain or frequency domain, and RRs are assessed based on these features. Finally, the evaluated RRs are combined to enhance prediction accuracy.

The long-term evolution of deep learning (DL) has propelled further advancements in respiratory rate prediction using ECG and PPG signals. Compared to signal-processing methods, DL-based approaches tend to predict RR in an end-to-end manner, utilizing the robust auto-feature-extraction capabilities of DL models. Ravichandran et al. [18] proposed RespNet employing a U-Net architecture [19], predicting RR from a given PPG signal with a mean squared error (MSE) of 0.262 on the CapnoBase dataset [20]. Leveraging a ResNet block, Bian et al. [21] trained another CNN-based (convolutional-neural-network-based) model on both synthetic and real data, yielding a mean absolute error (MAE) of 2.5. A lightweight model was introduced by Chowdhury et al. [22] employing a ConvMixter architecture to predict RR from PPG signals. This is also a CNN-based model and can be deployed on mobile devices for real-time monitoring. Compared to CNN-based models, RNN-based (recurrent-neural-network-based) models are adept at considering temporal relations more comprehensively. Baker et al. [23] constructed a bidirectional long short-term memory (BiLSTM) model to predict RR from the feature vectors extracted from ECG and PPG signals. Kumar et al. [24] compared seven models comprising CNN, LSTM, and their combinations, and their results indicated that LSTM and Bi-LSTM models with attention mechanisms exhibited superior accuracy in predicting RR from ECG and PPG signals.

Despite the considerable enhancement in RR prediction accuracy using DL-based models, their generalization capability has yet to be rigorously confirmed. DL-based models are trained using existing ECG and PPG data; however, ECG or PPG signals may vary among subjects and between patient groups [25], and significant inter-subject variations have been observed [13,26]. This makes the predictions of models trained on existing data unreliable when applied to new data. Specifically, modulations, such as FM, may be diminished in certain demographics, like the elderly [13]. These differences might also stem from variations in signal acquisition equipment and physicians’ operations. Moreover, the availability of labeled ECG or PPG data for RR is limited, with current data typically only drawn from a few hundred subjects. This scarcity of data could cause DL models to merely memorize input samples rather than truly extracting significant RR-related features, calling into question the generalization of DL-based methods. Many studies train and test their models on samples from the same subjects (different periods of time, i.e., segment-level), thereby raising doubt on the reliability of prediction accuracy when applied to a new subject. Additionally, to the best of our knowledge, very few DL-based methods provide the trained models or executable codes, making the application and comparison of these methods challenging.

In this study, we introduce a new model, TransRR, to predict RR utilizing ECG and PPG signals. Unlike conventional CNN- or RNN-based DL models, TransRR is built using the improved encoder of the Transformer model [27]. We trained and evaluated TransRR using different subjects (referred to as the subject-level approach) for reliably testing the model’s generalization performance. Additionally, we compared TransRR with the several popular DL-based RR prediction models. Our results demonstrated that TransRR surpassed all these models in MAE, percentage error (E), and Pearson correlation coefficient (PCC). Our processed data and Python codes are available at https://zenodo.org/record/7992684 (accessed on 10 August 2023) and https://github.com/mianfei71/TransRR/tree/main (accessed on 10 August 2023), respectively.

The remainder of this paper is organized as follows. Section 2 presents the materials and methods used in this study. Section 3 introduces the experiments conducted in this study and their results. Section 4 provides discussions on these results. Finally, Section 5 offers a conclusion of the entire paper.

## 2. Methods

### 2.1. Datasets

To facilitate comparisons with TransRR in subsequent studies, we exclusively utilized publicly available datasets in this study, instead of any synthetic or private data. We selected two publicly accessible datasets, BIDMC [28] and CapnoBase [20], to train and test TransRR. These datasets have also been employed in other DL-based RR prediction studies. Both datasets record ECG, PPG, and a simultaneous reference RR. Key details about these datasets are summarized in Table 1.

The BIDMC dataset, derived from the MIMIC-II resource [29], comprises ECG, PPG recordings, and simultaneous IP signals from 53 adult intensive care patients. Each recording is eight-minutes long, with a sampling rate of 125 Hz. The IP signals in each record were used to calculate reference RRs. Every breath cycle in the IP signals was manually annotated by two independent research assistants, and both sets of annotations were used to obtain the reference RR values.

The CapnoBase dataset encompasses ECG, PPG recordings, and capnography data, all of which are sampled at a frequency of 300 Hz. The subjects included in the dataset were randomly extracted from an extensive repository of physiological signals acquired during elective surgical procedures. The dataset includes a total of 42 subjects, each lasting for a duration of eight minutes, obtained from a cohort of 29 pediatric and 13 adult patients. These recordings were documented under both spontaneous and controlled breathing scenarios. Each breath cycle of the capnogram signals incorporated in the database was manually annotated by a research assistant. The time intervals between consecutive breaths were utilized to determine the reference RR values, based on these annotations.

### 2.2. Data Preprocessing

The data preprocessing pipeline is shown in Figure 1. Prior to preprocessing, we excluded data from one subject (bidmc_13) in the BIDMC dataset where over 75% of the RR label was missing. We also removed 10 subjects aged 5 or younger from the CapnoBase dataset, as their RR was significantly higher than the others (Figure 2). Following this, we resampled the ECG (lead 2) and PPG signals in both datasets to 125 Hz, and the RR to 1 Hz. This was performed to enable TransRR to be trained and tested on both the BIDMC and CapnoBase datasets. We then applied a 0.1~0.6 Hz band-pass filter (Butterworth Infinite Impulse Response Filter) to extract the respiratory signal from the ECG and PPG signals (this frequency range reflects the physiologically viable range of RR of 6~36 bpm). The extracted RR-related signals are often contaminated by motion affect (MA), which can hinder the DL model from learning meaningful features. To address this, we utilized variational mode decomposition (VMD), and eliminated the last mode out of the 5 modes extracted from the ECG and PPG signals. Generally, the last mode contains most of the MA [30]. In both BIDMC and CapnoBase datasets, inter-subject variation was considerable. Using such data, it is difficult to train a well-performed DL model. To mitigate inter-subject variability, z-score normalization was applied to the ECG and PPG signals. Subsequently, these signals were partitioned into 16 s windows with an 87.5% overlap. Finally, we removed the segments with RRs in the unreasonable range (equal to or smaller than 5 and equal to or larger than 30 bpm). In total, 11,900 and 6717 segments were obtained from the BIDMC and CapnoBase datasets, respectively.

### 2.3. Model Architecture

We designed TransRR (Figure 3) by utilizing the improved encoder of the Transformer [27] model. The Transformer model has significantly contributed to advancements in the field of natural language processing, and has found extensive application in temporal data processing [31,32]. We refined the structure of its encoder to adapt it to the RR prediction problem. As depicted in Figure 3, the architecture of TransRR is composed of three modules: the input module, the Transformer module, and the output module.

The input module takes ECG and PPG signals (down-sampled by a factor of one-eighth) with a length of 250 (16 s). A positional encoding, implemented through sine and cosine functions with varying frequencies as the Transformer model, is added to the ECG and PPG signals, providing the necessary positional information. Following this, the ECG and PPG signals with positional information are concatenated into a 250 × 2 matrix.

The Transformer module is the core of the TransRR model. This module is based on the encoder of Transformer, enhanced with two types of inception blocks [33]: kernel inception and dilation inception. TransRR initially employs multi-head self-attention to discern the internal relationships within the signals (head size = 32, number of heads = 8). The output of this process is then added with the input of the Transformer module through a residual connection and subsequently subjected to layer normalization. The normalized embeddings are next directed into a kernel inception block, which captures various local features through different sizes of convolution kernels (kernel size = 3, 5, 7, 9). To assimilate information in a larger receptive field and further integrate multi-scale contextual information, a dilation inception block is utilized with dilation rates of 1, 4, 8, and 16 (convolution kernel size = 5). These two inception blocks are followed by a dropout layer and a 1D CNN layer, respectively, to prevent overfitting and to adjust the number of channels. Finally, another residual connection and layer normalization are implemented. In the TransRR model, four Transformer modules are stacked for feature encoding.

The output module generates an RR prediction based on the encoded features. These features are flattened into a vector and subsequently fed into a DNN block. This block comprises four fully connected layers (FC), each followed by a dropout. The number of neurons for these FC layers is configured to 256, 64, 16, and 1. The output from the last FC layer outputs the predicted RR.

### 2.4. Model Performance Metrics

Upon generating predictions from the trained models on the test set, we assessed the performance of these models through the MAE, E, PCC, and 95% limits of agreement (LOAs), which have widely been adopted in similar RR prediction research. The calculation formulas of these metrics are presented in Table 2. The MAE and E evaluate the absolute error and relative error of predicted RRs relative to the reference RRs, respectively. The PCC quantifies the degree of linear correlation between the predicted RR and the reference RR, a measurement critical in real-world clinical applications. In instances of continuous monitoring, it is often of greater importance to track trends in RR as opposed to obtaining absolute values. LOA is an indicator of the agreement between the predicted RR and the reference RR. Specifically, the 95% LOA indicates that, with 95% confidence, the error of the predicted value falls within the LOA range. The computation of these metrics is summarized in Table 2.

## 3. Results

### 3.1. Overview of Data

In this study, we utilized two openly accessible datasets, BIDMC [28] and CapnoBase [20], with the differences between them detailed in Table 1. These data encompass 95 subjects of varying ages, genders, and health conditions. Figure 4 shows the distribution of ECG and PPG signal intensities for each subject in both datasets, respectively. F-tests indicated that the ECG and PPG distributions of different subjects in both datasets were significantly different (*p* << 0.001), and the inter-dataset variance was also significant (*p* << 0.001). These variations could come from differing sampling equipment, operator sampling methods, subjects, and environmental noise.

Inter-subject or inter-dataset variance poses a significant challenge for training a DL model. To mitigate these variances, we applied z-score normalization to the ECG and PPG signals from each subject in both datasets (refer to Materials and Methods). After normalization, the average and standard error across different subjects were standardized to 0 and 1, respectively, effectively removing inter-subject variance (*p* > 0.05). This also substantially reduced inter-dataset variance. Figure 5 shows UMAPs [34] for ECG (Figure 5a,b) and PPG (Figure 5c,d) signals before and after z-score normalization. It is evident that, prior to z-score normalization, the ECG and PPG signals from various subjects clustered according to the datasets. These clusters either merged (Figure 5b) or drew closer (Figure 5d) following z-score normalization. Nonetheless, the variance was not entirely eliminated (Figure 5d).

For RR, there were also significant distribution differences across different datasets (Figure 6, T-test, *p* << 0.001), although the means were closely matched. One of the reasons for the discrepancy was different age distributions within the two datasets (Table 1).

The raw ECG and PPG signals from each dataset were processed according to the pipeline outlined in Figure 1. Subsequently, the datasets were divided into ten subject-level folds for ten-fold cross-validation (nine folds for training, and one fold for testing). This approach ensures that no sample in the test set originated from the same subjects as those in the training set, providing a more robust evaluation of our model’s generalization capabilities. Within the training folds, 20% of the samples (subject-level) were allocated as the validation set.

### 3.2. Model Training

The TransRR model was trained on a workstation equipped with two CPUs (Xeon 8164) and two GPUs (GeForce 2080 RTX Ti). The model was trained for 100 epochs with a batch size of 128, with early stopping to prevent overfitting. This strategy stops the training process if there is no decrease in loss observed over 20 consecutive epochs. The MAE was utilized as the loss function, and Adam was selected as the optimizer with a dynamic learning rate ranging from 0.001 to 0.0001 (patience parameter was set to 5, with a decrease factor of 0.9). The performance of TransRR was evaluated using the subject-level ten-fold cross-validation approach, with metrics including MAE, E, PCC, and LOA (Table 2).

### 3.3. Model Performance

In order to evaluate our model’s generalization across different subjects within each dataset and across both datasets, we conducted five separate experiments. The first two experiments trained and tested TransRR using only the BIDMC or CapnoBase dataset. The first two columns of Table 3 show that the average MAEs (BIMDC: 1.33, CapnoBase: 0.96) and Es (both 0.08) obtained from subject-level ten-fold cross-validation were relatively low. Subsequently, we employed one of the two databases as the training set and the other as the test set, with the results presented in the third and fourth columns of Table 3. Both the average MAEs of the two experiments showed a significant increase, and the PCCs similarly decreased. These diminished performances can be attributed to the different sample distributions across the two datasets, which resulted in poor model generalization. This suggests that the performance of RR prediction models on a single dataset may not hold practical significance. Finally, we combined the training sets, validation sets, and test sets from both datasets. The average accuracy of TransRR on the combined test set is shown in the last column of Table 3. The average MAE of the subject-level ten-fold cross-validation fell between the values obtained in the BIDMC-only and CapnoBase-only circumstances. However, there was a noticeable increase in the PCC.

We carried out several ablation experiments to examine the impact of the two inception blocks (dilated inception and kernel inception) within the Transformer module of TransRR and the use of z-score in the data preprocessing pipeline. If any one of the two inception blocks was removed from TransRR, the output from the preceding layer was directly input into the subsequent layer. All experiments were performed using the combined dataset (CapnoBase + BIDMC), and subject-level ten-fold cross-validation was conducted in each case. The results are presented in Table 4. It can be seen that removing any component resulted in degraded performance for TransRR. When z-score was excluded from the preprocessing, the average MAE from cross-validation increased to 1.49. This suggests that normalizing raw ECG or PPG signals is a crucial step in enhancing the prediction performance of DL models. After the removal of the dilated inception block, the prediction accuracy significantly declined; however, it remained superior to the best of the other DL-based models used for comparison (Table 5 and Table 6). The highest MAE was obtained using TransRR without the kernel inception block (MAE = 1.91), and the E, PCC, and LOA were comparable to TransRR without the kernel inception block. These findings indicate that the inception blocks were beneficial for the TransRR model.

The performance of TransRR was evaluated in comparison with five popular DL-based RR prediction models known as of January 2023 (Table 5). Among them, RespNet [18], ResNet [35], and ConvMixer [22] models are based on CNN, while BiLSTM [23] and BiLSTM + ATT [24] models are based on RNN. In BiLSTM models [23], the input was the features of PPG and ECG signals, while the input for the other models was the raw ECG or PPG signals (additional attributes of these models are compared in Table 5). We reproduced these models, trained, and tested them using the same data (CapnoBase + BIDMC), and the same subject-level ten-fold cross-validation was conducted. The results are displayed in Table 6. Clearly, TransRR outperformed all five previous models in terms of MAE, E, and PCC. Specifically, TransRR reduced MAE by 36.5% compared to the lowest MAE (BiLSTM) achieved by the five models. However, the LOA for TransRR was looser than those for the BiLSTM model, particularly at the upper bound. These results suggest that TransRR possesses superior generalization performance compared to prior models.

## 4. Discussion

In this study, we proposed a novel RR prediction model, TransRR, utilizing the encoder of the Transformer. Utilizing ECG and PPG signals as inputs, TransRR demonstrated a high accuracy in RR prediction. To evaluate the generalization capacity of TransRR, we conducted a subject-level ten-fold cross-validation. On comparing TransRR with five popular DL-based RR prediction models, TransRR demonstrated superior performance in terms of MAE, E, and PCC.

In our work, we employed two datasets, BIDMC and CapnoBase, for the training and evaluation of the performance of TransRR. While previous studies have also employed multiple datasets, the significant variances within individual subjects and datasets were not thoroughly reported. In Section 3.1, we demonstrated that these variances were indeed significant. Such differences may stem from factors including, but not limited to, different sampling equipment, varied operator sampling modes, environmental noise, and the heterogeneity of subjects.

It is worth noting that PPG and ECG signals exhibit periodicity, and signal segments collected from a single subject are typically similar. Including segments from the same subject in both training and test sets can lead to considerable sample leakage, thereby potentially overestimating the model’s performance (in fact, the accuracy of the segment-level ten-fold cross-validation was significantly higher in all five compared models and TransRR). Yet, a model trained using such an approach may encounter difficulties when generalizing to new subjects.

To obtain a more reliable predictive accuracy for TransRR and the five previous models, we conducted subject-level ten-fold cross-validation. In alignment with our expectations, the accuracy of TransRR, when trained and tested using a single dataset, was exceptionally high. However, this performance notably declined when TransRR was trained and tested using different datasets (Table 3). This suggests that the RR prediction accuracy of a model trained and evaluated with a single dataset may lack clinical significance. We hypothesize that this degradation stems from the inherent bias amongst different datasets. Despite the fact that the performance of TransRR improved when employing both datasets for training and testing, ensuring the accuracy of predictions for a new subject remains challenging. In our opinion, in addition to the 95 subjects encompassed by the BIDMC and CapnoBase datasets, more data are required to train a highly generalizable and clinically applicable RR prediction model.

While TransRR surpasses the performance of other DL-based models, its accuracy remains limited. One primary factor may be the small size of the subject group available for model training (with fewer than 100 subjects involved). Although DL models are proficient at extracting features from a substantial quantity of samples, when the sample size (or number of subjects) is limited, these models struggle to acquire effective features. In such circumstances, a sophisticated data preprocessing strategy incorporating domain knowledge becomes crucial. In our study, we established a preprocessing pipeline (Figure 1) to minimize noises as much as possible. Specially, the first step of the pipeline involved excluding subjects aged five or younger, owing to the significant differences in respiratory patterns and generally high RRs compared to older subjects (Figure 2). Our findings (Table 4) also demonstrated the impact of the normalization step in the pipeline.

All the subjects used in our study were in a state of ill health (Table 1); especially, the subjects in BIDMC dataset were critically ill patients in intensive care units (ICUs). Their physiological signals can be significantly affected by ICU equipment, leading to distinct data distributions. This introduces substantial noise into the data, potentially impacting the performance of DL models. Despite the efforts made to mitigate the data noise, there remains a possibility that TransRR could be a biased model.

To the best of our knowledge, the majority of RR prediction models based on DL are principally based on CNN or RNN. However, CNN models may not adequately address temporal relationships, while RNN models possess limited capabilities in managing long-distance dependency issues. Unlike these models, the TransRR model fundamentally relies on the attention mechanism. This architecture can manage both temporal relationships and long-distance dependencies, thereby avoiding the aforementioned issues. The kernel inception integrated within the model facilitates the extraction of local features using variously sized kernels. Given the high sampling rate of ECG and PPG signals, we used two larger kernels (kernel size = 7, 9) to extract local features, a practice seldom adopted in image-related DL models. Further, we employed dilation inception to extend the receptive field. Our findings suggest that the improved architecture of TransRR substantially contributes to its superior performance.

Many previous studies have not provided sufficient transparency by withholding the processed training and testing data as well as the runnable codes, which significantly hinders the ability to conduct a meaningful horizontal comparison of model performance. In this study, we have addressed these issues by reproducing the compared models based on the information provided in the respective literature. Furthermore, we have trained and tested these models using the same dataset as TransRR. Although some details were omitted in their publications, we have made diligent efforts to optimize them. To foster reproducibility and facilitate the clinical application of RR prediction models, we have made all the preprocessed data and Python codes openly accessible, enabling others to replicate our work and conduct horizontal comparisons. We firmly believe that the dissemination of this information will contribute to advancing the field of RR prediction and its practical implementation in clinical settings.

## 5. Conclusions

This study introduces a novel predictive model, TransRR, based on the Transformer encoder, designed for estimating RR from ECG and PPG signals. The input signals underwent a sophisticated preprocessing pipeline, which incorporated domain knowledge. To assess the generalization capability of TransRR, a subject-level ten-fold cross-validation was conducted. When benchmarked against five popular DL-based models, TransRR demonstrated superior performance in the majority of the metrics. Despite these successes, the model necessitates exhaustive training across a diverse range of subjects, with an emphasis on healthy individuals. The current performance of TransRR (MAE: 1.20, E: 0.07, PCC: 0.85) surpassed those of five popular DL-based RR prediction models. However, it is still far from the requirement of clinical application. Future research should focus on further improving the prediction performance.

In the future, we believe more accurate models will emerge and provide valuable insights into the overall respiratory health of individuals. This advancement will result in an essential predictive tool that can be seamlessly integrated into various clinical settings, including ICUs, emergency departments, and general patient care. This integration will facilitate real-time assessments and prompt interventions in cases where deviations from normal respiratory rates are detected.

## Figures and Tables

**Figure 1 bioengineering-10-01024-f001:**
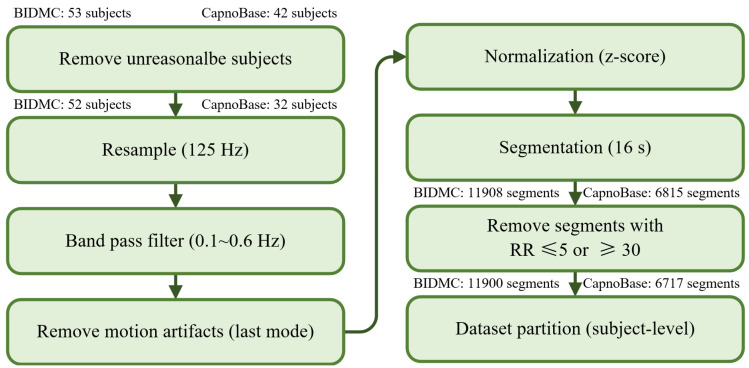
Data processing pipeline.

**Figure 2 bioengineering-10-01024-f002:**
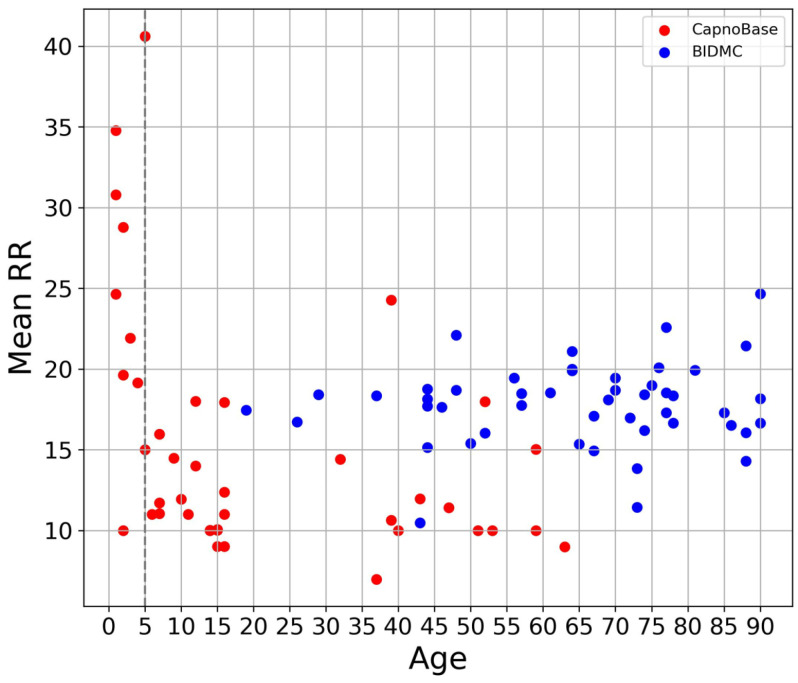
The distribution of average RR across varying age groups in BIDMC and CapnoBase datasets. Each data point corresponds to a subject, the x-axis denotes the subject’s age, and the y-axis denotes the average RR.

**Figure 3 bioengineering-10-01024-f003:**
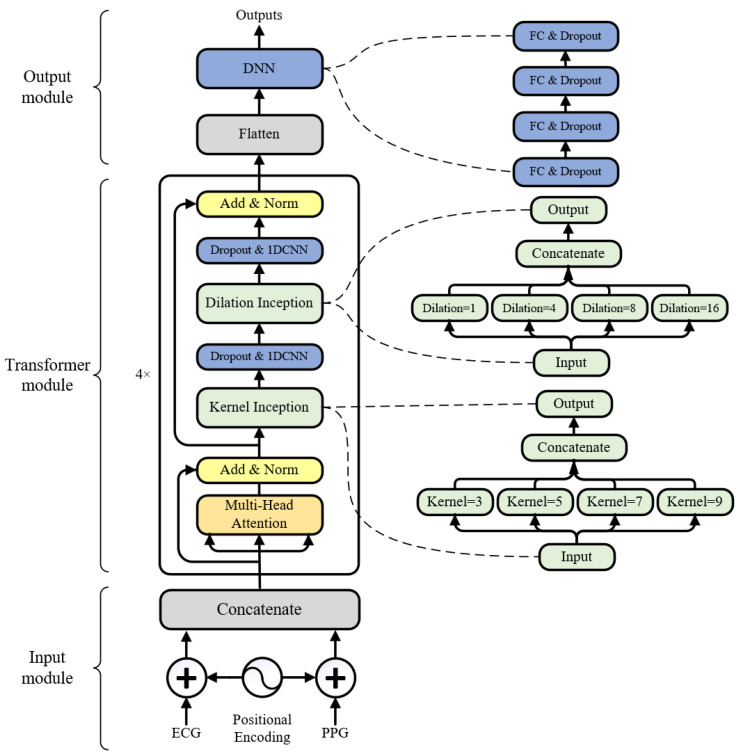
The proposed TransRR architecture.

**Figure 4 bioengineering-10-01024-f004:**
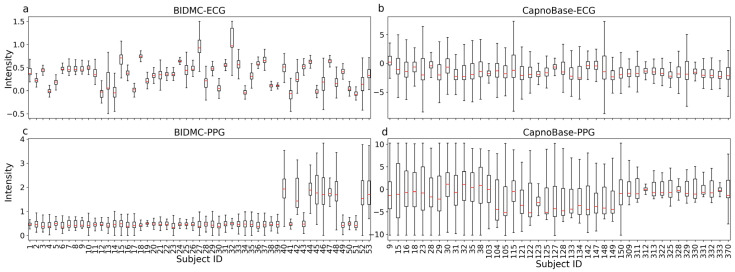
Significant inter-subject variations in ECG and PPG were observed across both BIDMC and CapnoBase datasets. The x-axis represents the subject ID, and the y-axis represents the signal intensity. Intensity values of each subject in both BIDMC and CapnoBase were used to draw the box plots.

**Figure 5 bioengineering-10-01024-f005:**
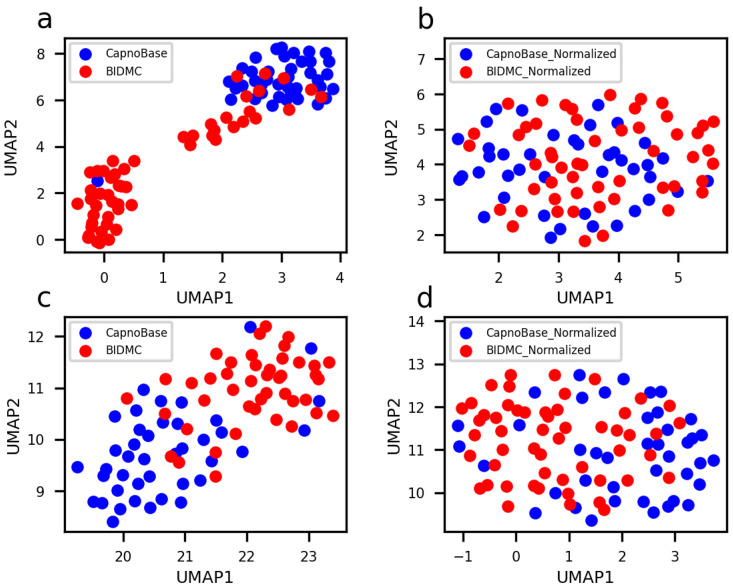
UMAPs for ECG (**a**,**b**) and PPG (**c**,**d**) signals prior to (**a**,**c**) and subsequent to (**b**,**d**) z-score normalization. Each point represents a single subject. The initial seven-minute segment of ECG or PPG signals were utilized as the features for each subject.

**Figure 6 bioengineering-10-01024-f006:**
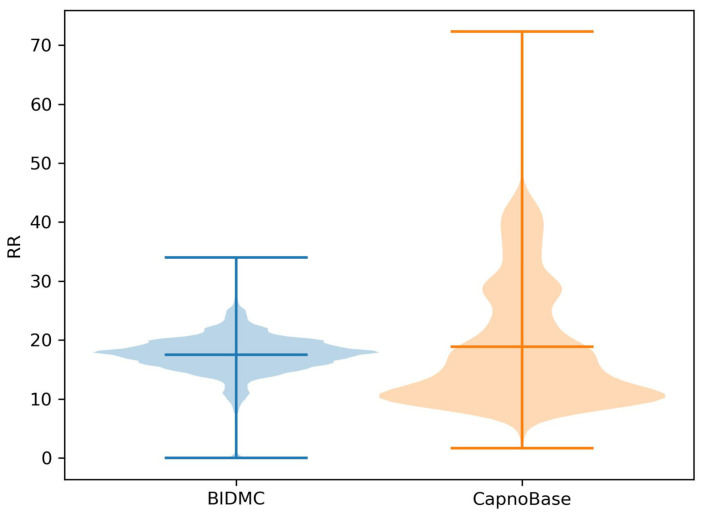
RR distributions comparison in BIDMC and CapnoBase datasets. The bars in the middle present the average of RR, and the bars above and below present the maximum and minimum values in both datasets, respectively.

**Table 1 bioengineering-10-01024-t001:** Summary of datasets.

Dataset	BIDMC [28]	CapnoBase [20]
Number of subjects	53	42
Duration	8 min	8 min
ECG/PPG sampling rate	125 Hz	300 Hz
RR sampling rate	1 Hz	n/a
The signal obtaining RR	IP	Capnometry
Male to female ratio	21:32	n/a
Children to adult ratio	0:53	29:13
Age range	18–88	0.8–75.6
Health condition	Critically ill	Selective surgery

**Table 2 bioengineering-10-01024-t002:** Model performance metrics.

Metric	Formula	Annotation
MAE	$\frac{1}{n}\sum_{i=1}^{n}\left|\hat{y_{i}}-y_{i}\right|$	n is the number of input samples. y and $\hat{y}$ are the reference RR and the predicted RR, respectively. μ is the mean of all reference RR. σ and $\hat{\sigma }$ are the standard error of reference RR and predicted RR, respectively. $\mu_{diff}$ and $\sigma_{diff}$ are the mean and standard error of the differences between y and $\hat{y}$, respectively.
E	$\frac{MAE}{\left|\mu \right|}\times 100$	
PCC	$\frac{cov(y,\hat{y})}{\sigma \hat{\sigma }}$	
LOA	$(\mu_{diff}-1.96\sigma_{diff},$ $\mu_{diff}+1.96\sigma_{diff})$	

**Table 3 bioengineering-10-01024-t003:** Performance comparison of TransRR trained on different datasets.

	Only BIDMC	Only CapnoBase	BIDMC→ CapnoBase ^a^	CapnoBase→ BIDMC ^a^	CapnoBase + BIDMC ^b^
MAE	1.33	0.96	3.47	3.98	1.20
E	0.08	0.08	0.29	0.23	0.07
PCC	0.61	0.63	0.23	0.19	0.85
LOA	[−3.46,3.71]	[−2.87, 3.11]	[−9.14, 5.99]	[−4.40, 11.13]	[−3.25, 3.97]

^a^ Model was trained on the dataset before “→”, and tested on the dataset after “→”. ^b^ Model was trained and tested using the combination of both dataset.

**Table 4 bioengineering-10-01024-t004:** Ablation study of TransRR on dataset CapnoBase + BIDMC.

	w/z-Score w/Inceptions	w/o z-Score	w/o Dilated Inception	w/o Kernel Inception
MAE	**1.20**	1.49	1.77	1.91
E	**0.07**	0.10	0.12	0.12
PCC	**0.85**	0.78	0.68	0.67
LOA	**[−3.25, 3.97]**	[−4.57,4.80]	[−5.12,5.57]	[−6.17,5.47]

Bold text represents the optimal result in each line.

**Table 5 bioengineering-10-01024-t005:** Comparison of five popular DL-based RR prediction models.

Model	Year	Model	Input	Output
RespNet	2019	Unet	PPG	RR
ResNet	2020	ResNet	PPG	RR
BiLSTM	2021	BiLSTM	The features of PPG and ECG	RR (max-min regulated)
BiLSTM + Att	2022	BiLSTM + Att	PPG and ECG	RR
Convmixer	2022	Convmixer	PPG	RR
TransRR (Ours)	2023	TransRR	PPG and ECG	RR

**Table 6 bioengineering-10-01024-t006:** Performance comparison of TransRR with five popular DL-based RR prediction methods.

	CNN-Based Models			RNN-Based Models		Transformer-Based Model
	RespNet	ResNet	ConvMixer	BiLSTM	BiLSTM + Att	TransRR (Ours)
MAE	3.76	3.06	2.73	** *1.89* **	2.81	**1.20**
E	0.25	0.20	0.17	** *0.15* **	0.34	**0.07**
PCC	0.32	0.44	0.43	** *0.46* **	0.17	**0.85**
LOA	[−9.61, 8.99]	[−8.32, 7.76]	[−6.92, 6.73]	**[−1.85, 3.72]**	[−5.04, 5.66]	** *[−3.25, 3.97]* **

Bold text represents the optimal result, and the italic text represents the suboptimal result in each line.

## Data Availability

Our processed data and Python codes are available at https://zenodo.org/record/7992684 (10 August 2023) and https://github.com/mianfei71/TransRR/tree/main (10 August 2023), respectively.

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
