# Peer review of "Predicting Respiratory Rate from Electrocardiogram and Photoplethysmogram Using a Transformer-Based Model"

_bioengineering, 2023, doi:10.3390/bioengineering10091024_

Round 1

Reviewer 1 Report

Nice paper with an  interesting approach ti assess RR from ECG combined with PPG.

According to your results a Pearson correlation coefficient of 85% nevertheless is rather low for clinical decision making. This aspect should be more clearly adressed in the conclusions.

References within the paper appear cannot be checked - all appear as "Error! Reference source not found"

Figures should be improved

Please find suggestions in the manuscript highlighted and commented in the sticky notes

minor errors

Reviewer 2 Report

references were not available.

overall this paper is suitable for this journal.

-

Reviewer 3 Report

I have reviewed the manuscript entitled ‘Predicting Respiratory Rate from Electrocardiogram and Photo- 2 plethysmogram Using a Transformer-Based Model’.

Please check the references according to the journal style.

The main aim of this study is to predict respiratory rate from ECG-based model. But the authors should explain well the potential role of this in their discussion

The manuscript is well-designed and written but the conclusions should be written more detailed.Artificial intelligence systems are widely being used in cardiology. The potential role of this system should be compared to other reported models citing ‘The Role of Artificial Intelligence in Coronary Artery Disease and Atrial Fibrillation’.

I have reviewed the manuscript entitled ‘Predicting Respiratory Rate from Electrocardiogram and Photo- 2 plethysmogram Using a Transformer-Based Model’.

Please check the references according to the journal style.

The main aim of this study is to predict respiratory rate from ECG-based model. But the authors should explain well the potential role of this in their discussion

The manuscript is well-designed and written but the conclusions should be written more detailed.Artificial intelligence systems are widely being used in cardiology. The potential role of this system should be compared to other reported models citing ‘The Role of Artificial Intelligence in Coronary Artery Disease and Atrial Fibrillation’.

Round 2

Reviewer 1 Report

paper has been revised in proper manner

minor errors in Grammar for instance "small" instead of "smaller" etc

Reviewer 3 Report

Thank you for the required revisions.